# Development of a plant-based dessert using araticum pulp and chickpea extract: Physicochemical, microbiological, antioxidant, and sensory characterization

Maria Carolina Mesquita[1]*, Márcio Antônio Mendonça[2], Raquel Braz Assunção Botelho[3], Sandra Fernandes Arruda[3], Eliana dos Santos Leandro[3]

1 Post-Graduate Program in Human Nutrition, Faculty of Health Sciences, Campus Universitário Darcy Ribeiro, University of Brasília, Brasília, DF, Brazil, 2 Faculty of Agronomy and Veterinary Medicine, University of Brasília, Brasília, DF, Brazil, 3 Department of Nutrition, Faculty of Health Sciences, Campus Universitário Darcy Ribeiro, University of Brasília, Brasília, DF, Brazil

* maria.carolinams@hotmail.com

**Data Availability Statement:** All relevant data are within the paper.

## Abstract

The demand for plant-based products has increased in recent years, due to several aspects related to health and environmental consciousness. This study aimed to produce and characterize a plant-based dairy alternative dessert based on araticum pulp and chickpea extract without added sugar and fat. Three formulations were prepared: Formulation 1 (F1): 20% araticum pulp + 80% chickpea extract; Formulation 2 (F2): 30% araticum pulp + 70% chickpea extract; and Formulation 3 (F3): 40% araticum pulp + 60% chickpea extract. All formulations' chemical composition, sensorial characteristics, viscosity, total phenolic content, antioxidant activity, and microbiological stability were analyzed during 28 days of storage at 4°C and a relative humidity of 23%. Energetic value ranged from 64 to 71 kcal/100g, and carbohydrate content from 9.68 to 11.06, protein from 3.38 to 3.04, lipids from 1.41 to 1.60, ashes from 0.53 to 0.59 and crude fiber from 0.86 to 1.34 g/100g among the formulations. The increase in the proportion of araticum pulp in the formulations reduced moisture content by 1.2 to 2.1% (F1: 84.2, F2: 83.2, and F3: 82.4), protein content by 3 to 9% (F1: 3.3, F2: 3.2, and F3: 3.0), and pH value by 5.8 to 10.7% (F1: 5.50, F2: 5.18, and F3: 4.91), and increased the TSS by 1.1 to 1.3-fold (F1: 8.36, F2: 8.98, and F3: 10.63 °Brix), total phenolics content by 1.5 to 2.0-fold (F1: 4,677, F2: 6,943, and F3: 10,112 gallic acid μmol/L) and antioxidant activity by 1.8 to 2.8-fold (F1: 1,974, F2: 3,664, and F3: 5.523). During the 28 days of storage at 4°C, the formulations F1 and F2 showed better stability of phenolic compounds and antioxidant activity; however, the formulation F3 showed acceptable microbiological quality up to 28 days of storage, higher viscosity, 8 to 16-fold higher than the formulations F1 and F2, respectively (F1: 238.90, F2: 474.30, and F3:3,959.77 mPa.s), antioxidant capacity and better scores in sensory analysis. The present study showed that the plant-based dessert elaborated with araticum pulp and chickpea extract might be considered a potential dairy alternative product with high antioxidant activity, protein content, and a viscosity similar to yogurt; however, its sensory aspects need improvement.

**Funding:** We thank Coordenação de Aperfeiçoamento de Pessoal de Ensino Superior (CAPES) of Brazil for the support with the scholarship (Maria Carolina Mesquita), as well as the financial backing provided by the Fundação de Support for Research in the Federal District (FAPDF) – Process: 00193-00000104/2019-86, (Eliana dos Santos Leandro).

**Competing interests:** The authors have declared that no competing interests exist.

## Introduction

The demand for plant-based products has increased in recent years. There is an inverse association between the consumption of plant-based products and the prevalence of chronic diseases [1]. Also, environmental consciousness, concern for sustainable food systems, the increasing number of people adept to veganism and vegetarianism and/or with allergies to milk proteins (α-lactalbumin, β-lactoglobulin, and casein) and intolerance to lactose [2] are some of the reasons that explain the current growing demand for plant-based food products.

According to the Plant-Based Food Association, the sales of plant-based dairy alternatives (PBDAs) rose by 9% in 2018, while those from cow milk fell by 6% [3]. In 2023, an increase of over 11% was expected for the overall market share of PBDA [4]. The EAT-Lancet Commission mentioned in 2018, that the consumption of fruits, vegetables, nuts, and legumes will have to double, while animal-source foods (meat and dairy) must be reduced by more than 50% by 2050. Both actions are essential to improve health and bring environmental benefits [5].

Fruits have been identified as the basis for the preparation of plant-based dairy alternative products [6–8], as they are sources of vitamins, minerals, fibers, and also of natural compounds that provide flavor and aroma [9]. In this context, the Brazilian Cerrado fruits may constitute an alternative ingredient for plant-based dairy alternatives. In addition to their nutrient content, Cerrado fruits have a high content of bioactive compounds with high antioxidant capacity [10, 11]. Phenolic compounds such as flavonoids, tannins, and anthocyanins represent the main bioactive compounds found in Cerrado fruits [12].

Among the fruits of the Brazilian Cerrado, araticum (*Annona crassiflora*) has unique sensory characteristics, attractive color, intense flavor, and exotic aroma, with high nutritional and technological potential. Araticum has a high content of phenolic compounds compared to other fruits, fibers and some polysaccharides with prebiotic properties [13–15].

Although fruits are good sources of carbohydrates, fibers, micronutrients, and bioactive compounds, they are poor in protein and lipids compared to dairy products [16, 17]. Therefore, the protein content constitutes a challenge in developing plant-based dairy alternatives. In this context, chickpeas may be considered an adequate plant-based protein alternative for plant-based dairy alternatives. Chickpeas contain protein, with essential amino acids such as lysine, leucine, phenylalanine, isoleucine and valine. It is considered the third most cultivated legume in the world and presents low allergenicity [18, 19]. The use of chickpeas as a food ingredient is currently poorly explored in the literature. Its properties and interaction with other food matrices are poorly understood [19].

Considering that the araticum fruit is rich in nutrients and non-nutrients, such as dietary fiber and phenolic compounds, and that chickpea is a source of proteins, including essential amino acids, this study hypothesizes that the interaction of these two products could result in a plant-based dessert alternative to dairy desserts. Thus, this study aimed to produce and characterize a plant-based dairy alternative dessert based on araticum pulp and chickpea extract without added sugar and fat.

## Materials and methods

### Araticum pulp preparation

Fifteen araticum (*Annona crassiflora*) fruits were purchased from a local street market in Brasília, Federal District, Brazil, and harvested from February to March 2022. The fruits were sanitized in a 2% sodium hypochlorite solution for 10 min, rinsed with drinking water, cut crosswise, and the carpels manually removed. The seeds were removed from the pulp, and the

pulp (9.6 kg of fresh weight pulp) was packed in plastic bags under vacuum and stored at–80˚C for 2 months until the beginning of the experiments.

## Chickpea extract preparation

A total of 7 kg chickpeas were purchased at a local market in Brasília, Federal District, Brazil (Bella Grão, Cristalina-GO, Brazil). Chickpea extract was prepared according to the process described by Rincon et al. [20]. Briefly, the raw grain was soaked for 12 h using drinking water at 4˚C, then the soaking water was discarded. The grains were cooked in water in a proportion of 1:3 under the following conditions: 2.0 atm at 120˚C for 20 min. The cooked grains were homogenized using a Thermomix® in the ratio of 1:4 (raw chickpea grains: water) and filtered on the voile to retain the residues.

## Formulations preparation

In order to determine the best percentage of araticum pulp and chickpea extract to produce a plant-based dessert alternative well-accepted by consumers, three formulations with different proportions of araticum pulp and chickpea extract were prepared: Formulation 1 (F1): 20% araticum pulp + 80% chickpea extract; Formulation 2 (F2): 30% araticum pulp + 70% chickpea extract; Formulation 3 (F3): 40% araticum pulp + 60% chickpea extract.

To prepare the formulations, araticum pulp and chickpea extract were homogenized in a blender (Philips Walita, model ProBlend 4) for 30 seconds at power one and then 30 seconds at power two. The homogeneous mixture was strained in voile to obtain the dessert. The preparation process of chickpea extract, araticum pulp, and formulations is summarized in a flowchart (Fig 1).

Aliquots of 400 mL of each formulation were placed in glass jars with screw caps and pasteurized at 65ºC for 15 minutes in a water bath (Warmnest, model HH-S10). After pasteurization, samples were cooled and stored at 4˚C for 28 days.

## Determination of chemical composition and energy value

The moisture content, protein, lipid, mineral residue, crude fiber, and total carbohydrates of formulations were analyzed. Furthermore, the energy value of the formulations was estimated using the Atwater general factor system that includes energy values of 4 kcal per gram (kcal/g) for protein, 4 kcal/g for carbohydrates, and 9 kcal/g for lipid. Moisture was determined according to the Analytical Procedures of Instituto Adolfo Lutz [21]. Protein content was determined according to the Kjeldahl 991.22 method [22]. The lipid content was determined according to the Am 5–04 extraction method [23]. Mineral residue was determined using method 945.45 [22]. Crude fiber content was determined according to method 978.10 [22]. The total carbohydrate amount was determined by difference, subtracting the values found for moisture, protein, lipid, and crude fiber from 100, according to the method 986.25 [22]. All analyses were assayed in triplicate.

## Stability of formulations during storage times

The physical-chemical and microbiological parameters of the different formulations F1, F2, and F3, were analyzed on days 1, 7, 14, 21, and 28 of storage at 4˚C and a relative humidity of 23%.

**pH and Total Soluble Solids (TSS).** The pH of the formulations was determined using a bench pHmeter (Digimed®, model DM21), using 10 g of sample, according to the Analytical Procedures of the Instituto Adolfo Lutz [21].

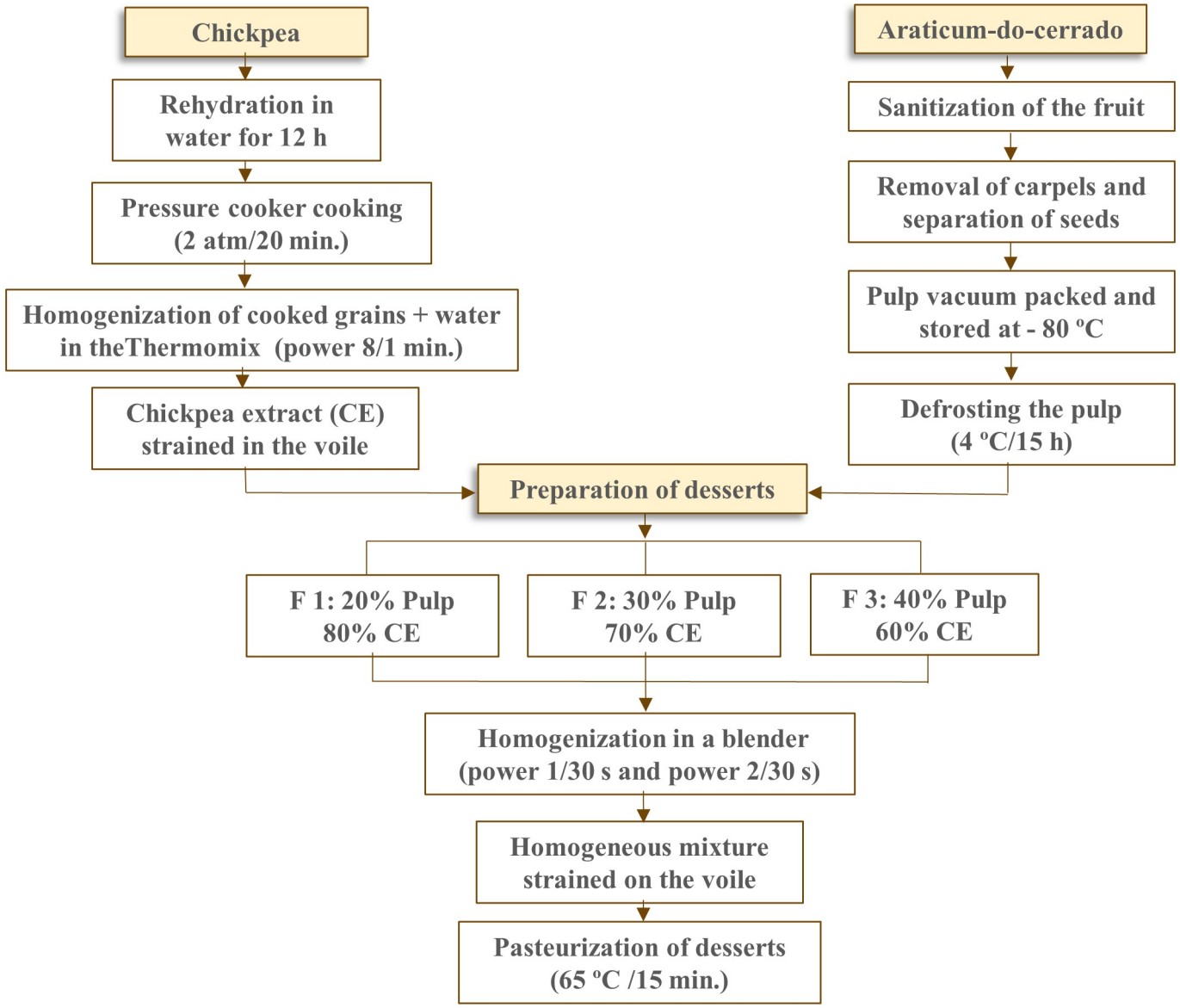

**Fig 1. Production flowchart of the formulations containing different proportions of araticum pulp and chickpea extract.**

Total Soluble Solids (TSS) were determined using a digital refractometer (ATAGO®, model 1 T), according to the method 932.12 [22], and the results were expressed in º Brix. All samples were assayed in triplicate.

**Microbiological analyzes.** The presence of mesophilic aerobic microorganisms, molds, and yeasts in the different formulations was analyzed according to the methodology described by the American Public Health Association [24]. Initially, the formulations were subjected to serial dilutions ($10^{-1}$, $10^{-2}$, and $10^{-3}$) in 0.85% (w/v) saline solution. The mesophilic aerobe microorganisms were determined by plating them in a standard agar culture medium (Neorgen®, Lansing, Michigan, USA). The plates were incubated in a bacteriological oven at 37°C for 48 hours, and the number of Colony Forming Units (CFU) was determined. The population of molds and yeasts was determined by plating them in a DRBC agar culture medium

(Neorgen®, Lansing, Michigan, USA). The plates were incubated in a bacteriological oven at 25˚C for five days, and the number of Colony Forming Units (CFU) was determined. Results were expressed in Colony Forming Units (CFU)/mL formulation. All samples were assayed in triplicate.

**Determination of total phenolic content and antioxidant activity.** *Sample preparation.* Sample extracts were prepared according to the protocol described by Larrauri et al. [25], with some modifications. Aliquots of 0.25 g of each sample were placed in microtubes, 2 mL of methanol/water (50:50 v/v) were added, and the reaction system remained under stirring for 1 hour at room temperature. Afterward, samples were centrifuged at 12,200 *xg* for 15 minutes at 25˚C, and the supernatant was collected. To the pellet, 2 mL of acetone/water (70:30 v/v) was added, and the previous procedure was performed. The methanol and acetone extracts were combined, and the final volume was adjusted to 5 mL with Mili-Q water to determine antioxidant activity and total phenolic content.

*Determination of total phenolic.* The determination of total phenolics was carried out according to the protocol described by Singleton and Rossi [26], which consists of the ability of phenolic compounds to reduce the mixture of Phosphomolybdic/Phosphotungstic acids in an alkaline medium. The determination of total phenolics was carried out in a 96-well plate, and the absorbance was measured at 765 nm, after 60 minutes of reaction under protection from light, in a spectrofluorimeter (SpectraMax M Series Multi-Mode Microplate Readers, San Jose, California). A standard curve of gallic acid was constructed at concentrations from 0 to 1,000 μmol/L, with a linear range of $R^2$ 0.99. The results were expressed as μmol gallic acid/L of the formulation, araticum pulp, or chickpea extract.

*Antioxidant activity.* The antioxidant activity of the samples was evaluated by assays of free radical scavenging capacity (2,2-diphenyl-1-picrylhydrazyl (DPPH•) and ferric reducing antioxidant potential (FRAP).

The ability to scavenge free radicals was determined using the radical 2,2-diphenyl-1-picrylhydrazyl (DPPH•), according to the protocol described by Cheng et al. [27]. The analysis was performed in a 96-well plate, and the absorbance was monitored at 515 nm every 30 seconds for 40 minutes in a spectrofluorimeter (SpectraMax M Series Multi-Mode Microplate Readers, San Jose, California). The antioxidant potential of the samples was calculated according to Eq 1. The results were expressed as mol TROLOX/L of the formulation, araticum pulp, or chickpea extract.

$$DRSC = \frac{AUC\ Sample}{AUC\ TROLOX} \times \frac{[Sample\ \mu L]}{[TROLOX\ \mu L]} \tag{1}$$

Where: DSRC is the relative DPPH• radical scavenging capacity; AUC is the area under the curve plotted from the variables "sample volume" x "% DPPH• scavenged".

The FRAP assay was performed according to the protocol described by Benzie and Strain [28]. The FRAP reagent was prepared using 0.3 mol/L acetate buffer (pH 3.6), 10 mmol/L 2,4,6-triazinetripyridyl in 40 mmol/L hydrochloric acid, and an aqueous solution of 20 mmol/L ferric chloride in a proportion of 10:1:1, and incubated at 37˚C for 30 min. The assay was performed by mixing 180 μL of FRAP reagent, 6 μL of sample extract, and 18 μL of deionized water. The analysis was performed in a 96-well plate, and the absorbance was measured at 593 nm, for 10 minutes of reaction in a spectrofluorimeter (SpectraMax M Series Multi-Mode Microplate Readers, San Jose, California). A standard curve was constructed with Trolox in a range of 0–1,500 μmol /L. The results were expressed as μmol TROLOX/L of the formulation, araticum pulp, or chickpea extract.

**Determination of viscosity.** To verify whether a higher proportion of araticum pulp could confer greater viscosity to the final product, the same proportions of araticum pulp used in the different formulations were homogenized in water and used as a control for the experiment: Control 1 (C1): 20% araticum pulp + 80% water; Control 2 (C2): 30% araticum pulp + 70% water; Control 3 (C3): 40% araticum pulp + 60% water. A natural whole yogurt (Itambé®, natural milk) was used as a viscosity standard.

The viscosity of the formulations and their controls was determined using a digital rotational viscometer Brookfield, model MVD-8 (Marte científica, Santa Rita do Sapucaí, MG, Brazil). Aliquots containing 10 g of each sample at room temperature (25 ºC) were used. The spindles #21, #22 and #23, were applied for 30 seconds, and the speed was set to 3, 6, or 12 rpm. The viscosity of the samples was analyzed at two storage times, days 1 and 28 at 4 ºC, and the results were expressed in mPa-s. All samples were assayed in triplicate [29, 30].

## Sensory analysis

Sensory analysis was conducted according to the guidelines established in the Declaration of Helsinki and approved by the Research Ethics Committee of the University of Brasília (nº 88754618.4.0000.0030/Emenda2/2023). The analysis took place at the Dietetic Technique Laboratory of the University of Brasilia, with 105 untrained participants recruited in a single day (May 31, 2023), who were randomly invited to participate in the research. Participants were instructed to read the Free and Informed Consent Form, and the sensory analysis started only after agreement and signature of the term. The inclusion criteria were: being 18 years or older and not having any allergy and/or intolerance to chickpeas and araticum.

Formulations F1, F2, and F3, described in item 2.3, were prepared and pasteurized one day before the sensorial analysis and submitted for evaluation by the participants. About 25 g of the formulations were placed in plastic cups and coded with three digits. Each participant received an empty cup to discard the formulations, a glass of water to drink between evaluations, a napkin, and disposable spoons. The formulations were presented randomly to the participants. They evaluated color, taste, odor, texture, and overall acceptance through a 9-point hedonic scale, 9 corresponding to "I liked extremely" and 1 "I disliked extremely" [31].

## Statistical analysis

A completely randomized design was adopted in a 3 x 5 factorial scheme, considering 3 formulations (F1, F2, and F3) and 5 time intervals (1, 7, 14, 21, and 28 days), except for the analysis of viscosity where a 6 x 2 factorial scheme was adopted, considering 6 formulations (C1, C2, C3, F1, F2, and F3) and 2 time intervals (1 and 28 days). All analyses were performed in triplicate. Data were expressed as mean ± standard deviation. Analysis of variance (ANOVA) was performed with subsequent Tukey's test, considering $P < 0.05$ for significant differences. A regression analysis was done for variables analyzed in the storage time intervals, such as pH and TSS. SAS software (SAS Institute Inc., Cary NC, version 9.4) was used for data analysis, and SigmaPlot software (Systat Software, Inc., Palo Alto, CA, version 14.0) was used to plot the graphs.

# Results and discussion

## Chemical composition

The chemical composition of the different formulations is described in Table 1. As expected, the moisture content (84.26–82.38 g/100 g) and protein (3.38–3.04 g/100 g) content of the formulations reduced as the percentage of araticum pulp in the formulations increased, as

Table 1. Chemical composition of the different formulations of plant-based dessert with araticum and chickpea extract.

| Formulations | Energetic Value (kcal/100 g) | Moisture | Carbohydrates | Proteins (g/100 g) | Lipids | Ashes | Crude fiber |
|---|---|---|---|---|---|---|---|
| F1 | 64 ± 0.66 | 84.26 ± 0.03[a] | 9.68 ± 0.15[b] | 3.38 ± 0.06[a] | 1.41 ± 0.18[a] | 0.53 ± 0.05[a] | 0.86 ± 0.00[b] |
| F2 | 69 ± 0.11 | 83.20 ± 0.06[b] | 10.72 ± 0.05[a] | 3.20 ± 0.05[b] | 1.42 ± 0.05[a] | 0.47 ± 0.04[a] | 0.93 ± 0.11[b] |
| F3 | 71 ± 0.06 | 82.38 ± 0.09[c] | 11.06 ± 0.06[a] | 3.04 ± 0.03[c] | 1.60 ± 0.02[a] | 0.59 ± 0.10[a] | 1.34 ± 0.05[a] |

Means followed by the same letter in a column do not differ statistically by the Tukey test ($P < 0.05$). n = 3. Formulation 1 (F1): 20% araticum pulp + 80% chickpea extract; Formulation 2 (F2): 30% araticum pulp + 70% chickpea extract; Formulation 3 (F3): 40% araticum pulp + 60% chickpea extract.

araticum pulp has lower moisture content (73.08 ± 0.05 g/100g) and protein content (1.37 ± 0.05 g/100g) compared to chickpea extract (93.7 ± 1.5 g/100g and 2.1 ± 0.07 g/100g, respectively) [32].

It is worth noting that the protein content observed in the formulations was similar to that obtained for yogurt and higher than other vegetable desserts based on soy, rice, coconut, macadamia, pea protein, and fava beans [33, 34]. The higher carbohydrate content in araticum pulp (18.92 ± 0.05 g/100g) compared to chickpea extract (3.39 ± 1.29 g/100g) [20, 32] resulted in an increase of this nutrient in formulations F2 (30% araticum pulp + 70% chickpea extract; 10.72 ± 0.05 g/100g) and F3 (40% araticum pulp + 60% chickpea extract; 11.06 ± 0.06 g/100g) in relation to formulation F1 (20% araticum pulp + 80% chickpea extract; 9.68 ± 0.15 g/100g). However, no significant difference was observed in carbohydrate concentration between formulations F3 and F2 (10.72 ± 0.05 and 11.06 ± 0.06 g/100g, respectively), despite the higher percentage of araticum pulp.

A similar result was observed for crude fiber content. The formulation F3 (40% araticum pulp + 60% chickpea extract) showed a higher content of this nutrient (1.34 ± 0.05 g/100g) than fomulations F1 (20% araticum pulp + 80% chickpea extract) and F2 (30% araticum pulp + 70% chickpea extract) (0.86 ± 0.00 and 0.93 ± 0.11 g/100g, respectively). No significant difference was observed in crude fiber content between F1 and F2, despite the higher percentage of araticum pulp. The fiber content of the formulations was similar to other desserts made from almonds, cashews, oats, macadamia, and rice [34].

Regarding energy value, similar results were obtained for the formulations, with values lower than 200 kcal/100 g. No significant differences ($P > 0.05$) were obtained for lipid and ash concentration among all formulations with the increase in the proportion of araticum pulp. Araticum pulp and chickpeas extract have low lipid content (2.39 ± 0.05 and 0.39 ± 0.22 g/100g, respectively), which may explain these results. The low calorie and lipid content of the formulations can be considered positive factors, as consumers seek plant-based dairy alternatives that are creamy and healthy, without excessive calories and fats [34].

## Stability during storage times

**pH and TSS.** The pH of the formulations, with different proportions of araticum pulp and chickpea extract, during 28 days of storage at 4 °C/ 23% humidity are presented in Fig 2 and Table 2. Araticum is considered an acidic fruit, with a pH around 4.0 [35, 36], therefore as the proportion of araticum pulp increased in the formulations (F1: 20%, F2: 30% and F3: 40% araticum pulp to chickpea extract F1: 80%, F2: 70% and F3: 60%), the average pH values decreased (5.50 ± 0.03, 5.18 ± 0.07 and 4.91 ± 0.09 for F1, F2 and F3, respectively), regardless of storage time. At day 7 of storage at 4°C/ 23% humidity, all formulations showed a significant decrease ($P < 0.05$) in pH values (5.55, 5.29, and 5.04 on day 1; 5.50, 5.15, and 4.85 on day 7,

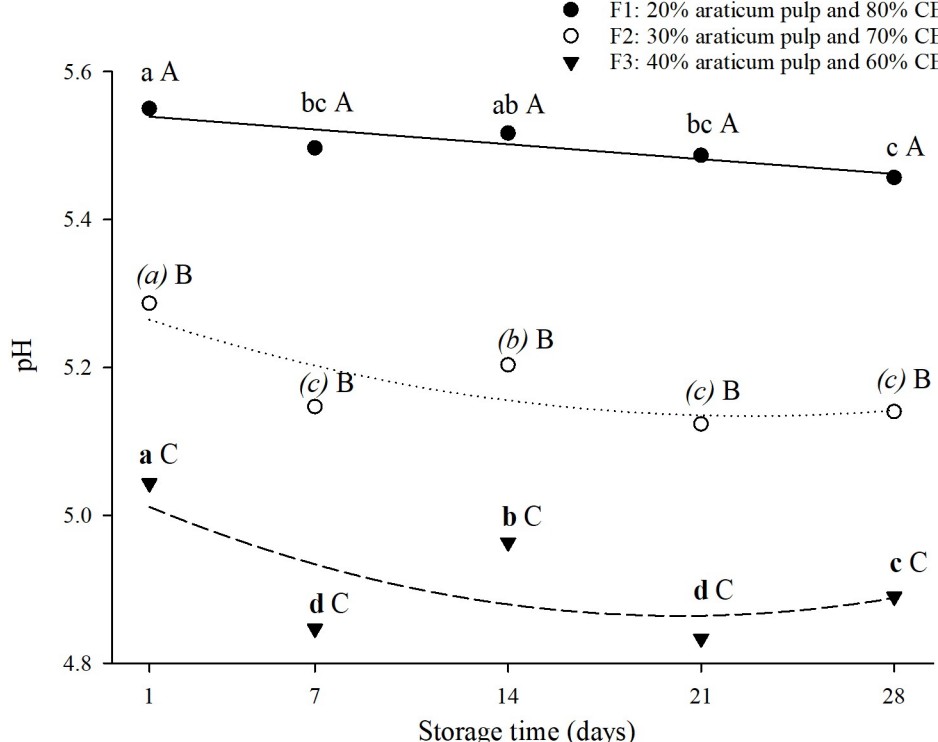

**Fig 2. pH regression curves of formulations F1, F2, and F3 during 28 days of storage at 4 ºC/ 23% humidity.**
CE = chickpea extract. Different capital letters within the same storage time indicate statistical differences between the formulations; different lowercase letters within the same formulation indicate statistical differences during the storage time by the Tukey test ($P < 0.05$).

for F1, F2, and F3, respectively; Fig 2). Formulation F1 (20% araticum pulp + 80% chickpea extract) maintained the pH value from day 7 to day 28 of storage. However, the formulations F2 (30% araticum pulp + 70% chickpea extract) and F3 (40% araticum pulp + 60% chickpea extract) presented a decrease from day 1 to 7, an increase from day 7 to 14, and a reduction in pH values from day 14 to 21.

The decrease in the pH values may be related to the degradation of components, such as polyphenols and organic acids during storage and the rapid conversion of proteins to amino acids, resulting in greater acidity [37]. Some studies have reported similar behavior for pH values in products developed with fruits [37–39]. The formulation F3 (40% araticum pulp + 60% chickpea extract) showed the lowest pH values ($P < 0.05$) during the storage time compared to

**Table 2. Regression equations of the pH and total soluble solids (TSS) during the storage period of the different formulations of plant-based desserts.**

| Variables | Formulation | Adjusted regressions | $R^2$ | SEE |
|---|---|---|---|---|
| **pH** | F1 | $\hat{y} = 5.5421 - 0.0029x$ | 0.79 | 0.018 |
| | F2 | $\hat{y} = 5.2772 - 0.0126 + 0.0003x^2$ | 0.66 | 0.055 |
| | F3 | $\hat{y} = 5.027 - 0.0161 + 0.0004x^2$ | 0.46 | 0.091 |
| **TSS** | | $\hat{y} = 10.0115 - 0.0969 + 0.0024x^2$ | 0.91 | 0.154 |

SEE = standard error of estimate; n = 3. Formulation 1 (F1): 20% araticum pulp + 80% chickpea extract; Formulation 2 (F2): 30% araticum pulp + 70% chickpea extract; Formulation 3 (F3): 40% araticum pulp + 60% chickpea extract.

formulations F1 (20% araticum pulp + 80% chickpea extract) and F2 (30% araticum pulp + 70% chickpea extract). The decrease in pH values in formulations can be considered a positive aspect since a lower pH value can inhibit the growth of pathogenic microorganisms [37].

The results of total soluble solids (TSS) demonstrates that there was a significant reduction ($P < 0.05$) in ºBrix values until day 14 of storage at 4˚C/ 23% humidity, regardless of the formulation composition. However, no significant difference was observed in TSS values from day 14 to day 28 of storage. (Fig 3A and Table 2). Regardless of the storage time, the increase of araticum pulp in formulations resulted in significantly ($P < 0.05$) higher TSS medium values (8.36 ± 0.02, 8.98 ± 0.31, and 10.63 ± 0.39 º Brix for fomulations F1 (20% araticum), F2 (30% araticum), and F3 (40% araticum), respectively; Fig 3B).

TSS is a group of substances that can be dissolved in water, including sugar, organic acid and protein. The decrease in TSS may be related to chemical reactions during storage, making them less soluble [40, 41]. The decrease in pH values during storage time reinforces this hypothesis. Since formulations were only pasteurized, some oxidative enzymes may not have been completely inactivated could and catalyze the oxidation of organic elements (polyphenols and organic acids) in the formulations, decreasing pH and TSS. These reactions can convert the soluble components in insoluble ones [42, 43]. Our results are in accordance with Namet et al. [42] that demonstrated a decrease in pH and an increase in acidity and total soluble sugars throughout storage in a functional drink made with melon by-product. Other studies have reported similar behavior for TSS in fruit products [38, 44].

The increase in araticum pulp in the formulations promoted an increase in TSS, and the formulations F2 (30% araticum pulp + 70% chickpea extract) and F3 (40% araticum pulp + 60% chickpea extract) showed a significantly higher TSS ($P < 0.05$) when compared to formulation F1 (20% araticum pulp + 80% chickpea extract). Araticum pulp has a lower moisture content (73%) [32] than chickpea extract (93%) [20], which may explain the increment in total soluble solids as its proportion in formulation increases. Therefore, the lower moisture content in F2 and F3 formulations resulted in higher TSS.

**Microbiological analysis.** The count of mesophilic aerobic microorganisms, molds, and yeasts in the three formulations at days 1, 7, 14, 21, and 28 of storage at 4 ºC/ 23% humidity is described in Table 3. The formulation F3 (40% araticum pulp + 60% chickpea extract), which has the higher proportion of araticum pulp, showed a low count of mesophilic aerobic microorganisms, molds, and yeasts ($< 10$ CFU/mL) during all 28 days of storage at 4 ºC/ 23% humidity. The formulations F1 (20% araticum pulp + 80% chickpea extract) and F2 (30% araticum pulp + 70% chickpea extract) presented a similar profile concerning mesophilic aerobic microorganisms, molds, and yeast count. From day 14 to day 28 of storage at 4 ºC/ 23% humidity, both formulations showed exponential growth of all studied microorganisms (F1: $4.3 \times 10^2$ to $1.3 \times 10^6$; F2: $1.4 \times 10^2$ to $4.2 \times 10^5$ CFU/mL for mesophilic aerobic count and F1: $3.2 \times 10^2$ to $1.6 \times 10^5$; F2: $3.1 \times 10^2$ to $1.2 \times 10^5$ CFU/mL for molds and yeasts).

Mesophilic aerobic, molds and yeasts are microorganisms that indicate the microbiological quality of food. Formulations F1 (20% araticum pulp + 80% chickpea extract) and F2 (30% araticum pulp + 70% chickpea extract) presented acceptable results for up to 21 days of storage at 4 ºC 23% humidity, as counts greater than $10^4$ of the microorganisms analyzed are considered unsatisfactory with unacceptable quality [45, 46]. However, only the formulation F3 achieved acceptable quality results during 28 days of storage at 4 ºC/ 23% humidity. It should be emphasized that none of the formulations were added of food additives to prevent the growth of microorganisms and thus extend the shelf life of desserts.

The low growth of these microorganisms in the formulations, especially in F3 (40% araticum pulp + 60% chickpea extract), may be related to the decrease in pH obtained by adding araticum pulp in the formulations. It is important to highlight that even though the pH of the three

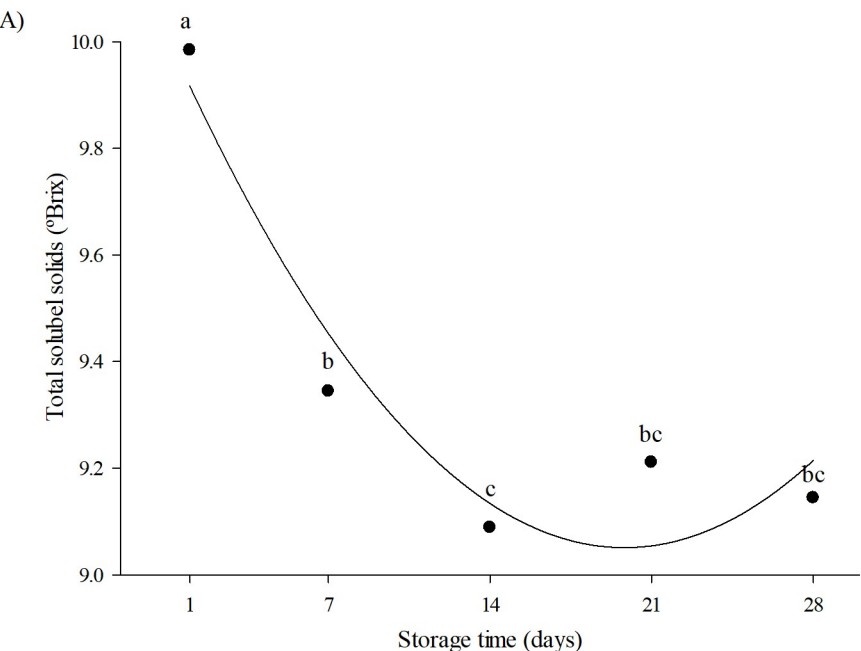

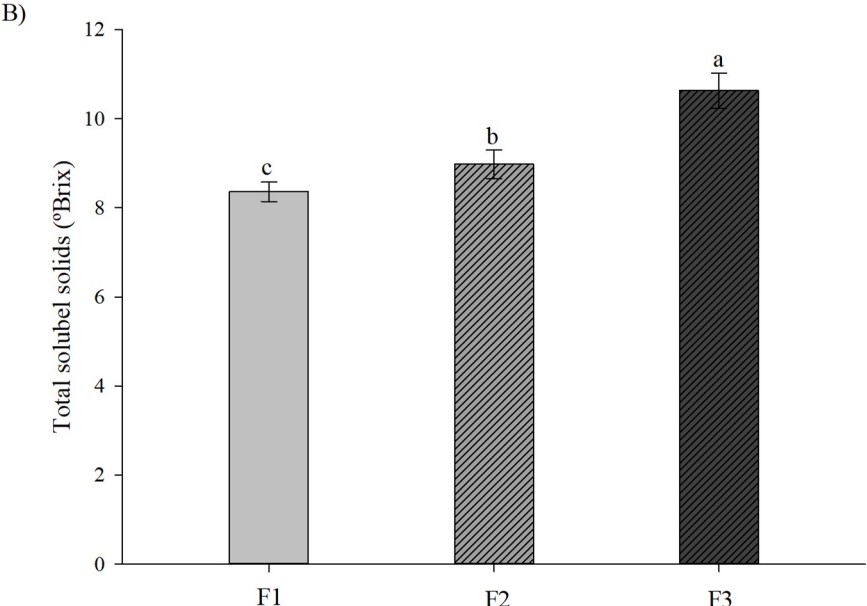

**Fig 3. Total soluble solids (ºBrix) values of the formulations F1, F2, and F3.** (A) Total soluble solids (ºBrix) values during storage time, independent of the proportion of araticum pulp and chickpea extract used in the formulations. (B) Total soluble solids (ºBrix) values of different formulations independent of the storage time. Formulation 1 (F1): 20% araticum pulp + 80% chickpea extract; Formulation 2 (F2): 30% araticum pulp + 70% chickpea extract; Formulation 3 (F3): 40% araticum pulp + 60% chickpea extract. Different letters indicate statistical differences by the Tukey test ($P < 0.05$).

**Table 3. Mesophilic aerobic bacteria, molds, and yeasts count (CFU/mL) in the different formulations of plant-based desserts during 28 days of storage at 4oC.**

| Storage time | Formulations | | |
|:---:|:---:|:---:|:---:|
| (days) | F1 | F2 | F3 |
| | Mesophilic aerobes bacteria (CFU/mL) | | |
| 1 | $< 10^*$ | $< 10^*$ | $< 10^*$ |
| 7 | $4.0 \times 10^2$ | $< 10^*$ | $< 10^*$ |
| 14 | $4.3 \times 10^2$ | $1.4 \times 10^2$ | $< 10^*$ |
| 21 | $1.4 \times 10^3$ | $1.1 \times 10^3$ | $< 10^*$ |
| 28 | $1.3 \times 10^6$ | $4.2 \times 10^5$ | $< 10^*$ |
| | Molds and yeasts (CFU/mL) | | |
| 1 | $< 10^*$ | $< 10^*$ | $< 10^*$ |
| 7 | $< 10^*$ | $< 10^*$ | $< 10^*$ |
| 14 | $3.2 \times 10^2$ | $3.1 \times 10^2$ | $< 10^*$ |
| 21 | $5.0 \times 10^3$ | $6.2 \times 10^3$ | $< 10^*$ |
| 28 | $1.6 \times 10^5$ | $1.2 \times 10^5$ | $< 10^*$ |

Formulation 1 (F1): 20% araticum pulp + 80% chickpea extract, Formulation 2 (F2): 30% araticum pulp + 70% chickpea extract, and Formulation 3 (F3): 40% araticum pulp + 60% chickpea extract. *Estimated values. n = 3.

formulations is close to the ideal pH for the growth of molds and yeasts, 4–6, a low count of mesophilic aerobic microorganisms, molds, and yeasts was obtained, suggesting that some compounds in araticum pulp have antimicrobial activity against the studied microorganisms [47].

Plants from the *Annonaceae* family, such as araticum, have secondary metabolites, such as alkaloids, acetogenins, and phenolic compounds with antibacterial activity [48]. Da Silva et al. [49], detected alkaloids in araticum pulp and attributed the antimicrobial activity against *S. aureus* ATCC 6538 to these compounds. Ramos et al. [50], when analyzing the araticum fruit, detected acetogenins, such as anonacinone and squamolinone, in the pulp and cited that these compounds presented antibacterial activity. Stafussa et al. [51], identified some phenolic compounds, such as quinic acid and catechin, in araticum pulp and attributed the antibacterial activity of araticum pulp against *S. aureus* and *E. coli* to these compounds. Therefore, the higher proportion of araticum pulp and antimicrobial compounds in the formulation F3 (40% araticum pulp + 60% chickpea extract) may explain the lower growth of microorganisms compared to the other formulations.

The antimicrobial activity of the compounds cited above is associated with the hydroxyl groups present in their chemical structures, which may destabilize bacterial cell membrane and inhibit bacterial enzymes, delaying the growth of microorganisms [52].

**Total phenolic content.** The total phenolic content of araticum pulp, chickpea extract, and different formulations during the storage time of 28 days at 4˚C/ 23% humidity is shown in Table 4. Araticum pulp had higher total phenolic content (22,796 ± 729 μmol gallic acid/L) compared to chickpea extract (1,456 ± 713 μmol gallic acid/L, Table 4). As expected, the total phenolic content of the three formulations increased proportionally to the concentration of araticum pulp. The formulation F3 (40% araticum pulp + 60% chickpea extract) presented the higher phenolic content (10,112 ± 250 gallic acid μmol/L) compared to formulations F1 (20% araticum pulp+ 80% chickpea extract) and F2 (30% araticum pulp + 70% chickpea extract) (F1: 4,677 ± 365 and F2: 6,943 ± 474 gallic acid μmol/L, respectively). These values were higher than those obtained for dairy and soy milk desserts, 1,970 to 4,560 gallic acid μmol/L [53], which demonstrates the potential of formulations made with araticum pulp and chickpea extract.

**Table 4. Total phenolic content (μmol/L) and antioxidant activity estimated by DPPH• scavenging capacity (mol/L) and by FRAP (μmol/L) assays of araticum pulp, chickpea extract, and different formulations of plant-based desserts during 28 days of storage at 4°C/ 23% humidity.**

| Storage time (days) | Samples | | | | |
|---|---|---|---|---|---|
| | Araticum Pulp | Chickpea extract | F1 | F2 | F3 |
| **Total phenolic (μmol/L)** | | | | | |
| 1 | 22,796.3 ± 729.7 [A] | 1,456.0 ± 713.1 [A] | 4,677.0 ± 365.2 [c;A] | 6,943.8 ± 474.8 [b;A] | 10,112.0 ± 250.2 [a;A] |
| 7 | 18,984.3 ± 590.0 [B] | 1,994.0 ± 143.8 [A] | 4,045.8 ± 275.3 [c;B] | 5,984.7 ± 327.0 [b;B] | 8,667.8 ± 477.3 [a;BC] |
| 14 | 17,541.0 ± 1386.7 [C] | 1,444.0 ± 69.6 [A] | 4,311.3 ± 128.0 [c;AB] | 6,176.3 ± 362.0 [b;B] | 9,043.5 ± 288.3 [a;B] |
| 21 | 19,255.0 ± 168.1 [B] | 1,820.8 ± 672.2 [A] | 4,107.2 ± 210.7 [c;AB] | 5,810.3 ± 364.0 [b;B] | 8,009.8 ± 389.5 [a;D] |
| 28 | 18,389.0 ± 1775.4 [B] | 1,646.7 ± 240.5 [A] | 3,948.7 ± 84.5 [c;B] | 5,617.0 ± 145.8 [b;B] | 8,220.0 ± 34.7 [a;CD] |
| **DPPH• (mol/L)** | | | | | |
| 1 | 64.6 ± 11.7 [A] | 3.1 ± 0.4 [A] | 15.1 ± 4.9 [b;A] | 26.9 ± 3.0 [ab;A] | 41.2 ± 5.4 [a;A] |
| 7 | 46.8 ± 3.9 [B] | 2.8 ± 0.5 [A] | 13.5 ± 5.6 [b;A] | 25.6 ± 1.2 [ab;A] | 34.1 ± 3.9 [a;B] |
| 14 | 44.4 ± 0.6 [B] | 3.1 ± 0.5 [A] | 13.7 ± 6.1 [b;A] | 26.1 ± 1.2 [ab;A] | 34.4 ± 3.7 [a;B] |
| 21 | 49.9 ± 8.8 [B] | 3.8 ± 0.5 [A] | 16.0 ± 5.2 [a;A] | 25.3 ± 2.6 [a;A] | 32.8 ± 5.1 [a;B] |
| 28 | 48.9 ± 6.1 [B] | 4.1 ± 0.5 [A] | 16.6 ± 5.5 [a;A] | 25.2 ± 1.6 [a;A] | 33.6 ± 4.7 [a;B] |
| **FRAP (μmol/L)** | | | | | |
| 1 | 18,970.0 ± 363.9 [A] | 1,467.8 ± 39.5 [A] | 1,974.4 ± 276.6 [c;A] | 3,664.3 ± 338.8 [b;A] | 5,523.6 ± 1,006.5 [a;B] |
| 7 | 13,514.0 ± 1187.8 [B] | 1,145.0 ± 105.7 [A] | 2,289.8 ± 102.9 [c;A] | 4,342.4 ± 303.5 [b;A] | 7,201.7 ± 498.4 [a;A] |
| 14 | 12,579.0 ± 708.6 [B] | 918.1 ± 65.0 [A] | 2,611.4 ± 215.8 [c;A] | 4,921.4 ± 438.9 [b;A] | 7,812.1 ± 317.1 [a;A] |
| 21 | 12,889.0 ± 1598.1 [B] | 906.7 ± 25.5 [A] | 2,781.0 ± 958.2 [b;A] | 4,151.9 ± 315.7 [b;A] | 6,333.6 ± 310.5 [a;AB] |
| 28 | 13,217.1 ± 1598.2 [B] | 871.4 ± 40.4 [A] | 2,198.8 ± 134.0 [c;A] | 3,897.4 ± 406.5 [b;A] | 6,504.5 ± 165.0 [a;AB] |

Means followed by the same lowercase letter in a row or the same capital letter in a column do not differ statistically by the Tukey test ($P < 0.05$). n = 3. Formulation 1 (F1): 20% araticum pulp + 80% chickpea extract, Formulation 2 (F2): 30% araticum pulp + 70% chickpea extract, and Formulation 3 (F3): 40% araticum pulp + 60% chickpea extract.

Vasco, Ruales & Kamal-Eldin [54], when categorizing fruits according to their phenolic content, suggested that those fruits with a phenolic content greater than 500 mg GAE/100 g fresh weight could be considered fruits with high levels of these compounds. Therefore, araticum pulp could be considered a fruit source of phenolic compounds. Arruda et al. [52], identified 139 phytochemicals in araticum pulp, of which 82 were phenolic compounds and their derivatives, representing 59% of all the identified phytochemicals. Some identified phenolic compounds were procyanidin, catechin, epicatechin, flavanomaein, quecetin, kaempferol, rutin, and caffeic acid. In addition to their antioxidant and antibacterial activities, the phenolic compounds have antitumor, anti-inflammatory, antihypertensive, and hepatoprotective properties [36, 55].

In the chickpea extract, the values of total phenolic content remained constant during the storage time of 28 days at 4°C/ 23% humidity. On day 7 of storage at 4°C, the phenolic content in the three formulations decreased compared to day 1. In the formulations F1 (20% araticum pulp + 80% chickpea extract) and F2 (30% araticum pulp + 70% chickpea extract), no difference was observed from day 7 to day 28 of storage in total phenolic content; however, in the formulation F3 (40% araticum pulp 60% chickpea extract), a significant decrease in phenolic content was observed on day 21 of storage compared to day 7.

These results suggest that the formulations with lower percentages of araticum pulp and consequently higher percentages of chickpea extract, F1 (20% araticum pulp + 80% chickpea extract) and F2 (30% araticum pulp + 70% chickpea extract), showed better stability of phenolic compounds during the 28 days of storage at 4°C/ 23% humidity. The better stability of

phenolic compounds in formulations F1 and F2 may be explained by the higher concentration of proteins, as phenolics interact with proteins. According to Guan et al. [56], phenolic compounds can interact with the peptide chains of vegetable and/or animal proteins through covalent and non-covalent bonds, and the conjugated complexes exhibit better stability, which preserves the antioxidant activity.

The presence of these amino acids in chickpea protein [57] reinforces the hypotheses that interactions between chickpea proteins and/or amino acids and the phenolic compounds of araticum pulp may have improved the stability of phenolic components during the storage of formulations with a higher percentage of chickpea extract.

The lower stability of total phenolics in formulation F3 (40% araticum pulp + 60% chickpea extract) throughout storage at 4ºC/ 23% humidity may be related to the decrease in the pH value during storage, which may diminish the stability of phenolic compounds. According to Ozdal Capanoglu and Filiz Altay [58], one of the main parameters that can affect the phenol-protein interaction is pH, as high values or values below neutrality ($< 7$) interfere with the degree of bonding between molecules. A pH closer to the protein and/or amino acid's isoelectric point would be ideal for a stronger bond.

**Antioxidant activity (DPPH• and FRAP).**   The antioxidant activity of araticum pulp, chickpea extract, and different formulations during storage at 4°C for 28 days/ 23% humidity is shown in Table 4. Araticum pulp showed higher antioxidant potential in both DPPH• and FRAP assays compared to chickpea extract (64.6 ± 11.7 *versus* 3.1 ± 0.4 mol/L and 18,970.0 ± 363.9 *versus* 1,467.8 ± 39.5 µmol/L; Table 4). The antioxidant activity of araticum pulp significantly decreased from day 1 to 7 of storage at 4°C/ 23% humidity, showing stability from day 7 of storage onwards. No significant differences were observed in the antioxidant potential of chickpea extract during storage time. The greater antioxidant activity in araticum pulp can be attributed to its higher total phenolic content when compared to chickpea extract, as mentioned previously.

The similarity between the results regarding total phenolic content and antioxidant capacity can be explained by the biological antioxidant activity exerted by phenolic species. Thus, a food's antioxidant potential can be attributed mainly to its total phenolic content [56].

The antioxidant activity of the different formulations estimated by the FRAP assay increased proportionally to the added content of araticum pulp. However, according to the DPPH• assay, the antioxidant activity of the formulations F1 (20% araticum pulp + 80% chickpea extract) and F2 (30% araticum pulp + 70% chickpea extract) was not significantly different ($P > 0.05$). During the 28 days of storage, the antioxidant capacity of formulations F1 and F2 remained constant, which reinforces the hypothesis of a possible interaction between phenolic compounds of araticum pulp and proteins from chickpea extract, which results in a better stability of the antioxidant activity of the formulations throughout storage at 4 ºC/ 23% humidity. Contrary to that observed for F1 and F2, the formulation F3 (40% araticum pulp + 60% chickpea extract), showed an increase in antioxidant potential from day 1 to 7 by the FRAP assay and a decrease by the DPPH• assay, followed by stability in values from day 7 to 28 of storage, by both assays.

The difference in the antioxidant capacity measured by the FRAP and DPPH• methods can be explained by the differences in the reactivity of various antioxidants in the chemical reaction of each method. FRAP method is based on the ability to reduce ferric tripyridyltriazine complex (Fe (III)-TPTZ) to blue ferrous complex (Fe (II)-TPTZ) by electron-donating antioxidants transfer, therefore measuring the reduction of the ferric ion. DPPH• method is based on the scavenging of the 1,1-diphenyl-2-picrylhydrazyl free radical and, therefore, it measures the radical neutralization capacity of antioxidants [59].

In the present study, the antioxidant capacity measured by the FRAP method seems to be correlated with the total phenolic content of formulations, and both are proportional to the percentage of added araticum pulp to the formulations. Thaipong et al. [60], reported that the antioxidant activity estimated by FRAP assay had a higher correlation with phenolic content than that obtained using other assays such as DPPH•, ABTS•, and ORAC.

In general, when observing the results of antioxidant capacity measured by the FRAP, the different formulations exhibit superior results compared to those obtained for dairy and soy milk desserts, ranging from 640 to 1,290 µmol/L [53]. Although formulation F3 (40% araticum pulp + 60% chickpea extract) showed lower stability in the first seven days of storage at 4ºC, its higher total phenolic content and greater antioxidant activity demonstrate the functional potential of the araticum and chickpea extract vegetarian dessert. It is well known that antioxidants can neutralize free radicals/ROS, preventing various chronic diseases and consequently promoting health [59, 61].

**Viscosity.** The viscosity of the different formulations and respective controls are presented in Fig 4. Formulation F3 (40% araticum pulp + 60% chickpea extract) presented the highest viscosity (3,959.77 mPa.s) compared with F1 (20% araticum pulp + 80% chickpea extract; 238.90 mPa.s) and F2 (20% araticum pulp + 80% chickpea extract; 474.30 mPa.s). A similar result was obtained for its control C3 (40% araticum pulp + 60% water). Although the formulation F2 has 1.5-fold more araticum pulp than the formulation F1, no difference was observed in viscosity between these formulations (474.30 and 238.90 mPa.s, respectively). These results agree with that obtained for their respective controls; C2's (30% araticum pulp + 60% water) viscosity was similar to C1's (20% araticum pulp + 60% water). Regardless of the percentage of araticum pulp, the viscosity of formulations did not alter after 28 days of storage.

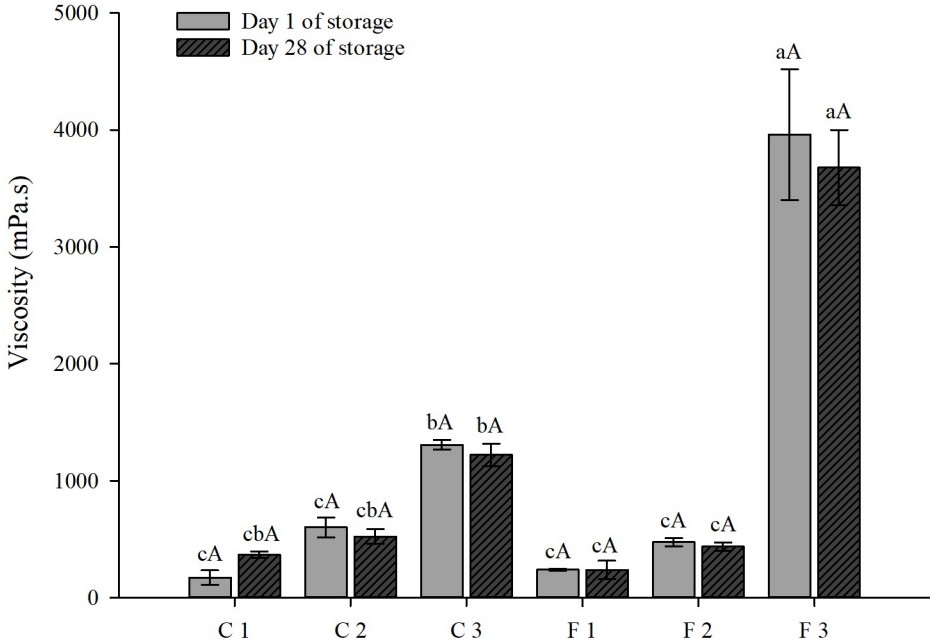

**Fig 4. Viscosity (mPa-s) of controls and test formulations on days 1 and 28 of storage at 4 ºC/ 23% humidity.** C1: 20% araticum pulp + 80% water, C2: 30% araticum pulp + 70% water, C3: 40% araticum pulp + 60% water, F1: 20% araticum pulp + 80% chickpea extract, F2: 30% araticum pulp + 70% chickpea extract, F3: 40% araticum pulp + 60% chickpea extract. Different lowercase letters indicate statistical differences between formulations and controls; different capital letters indicate differences in the storage time within the same formulation by the Tukey test ($P < 0.05$).

Although the addition of 40% araticum pulp to 60% water increased the viscosity of the mixture (1,307.43 mPa.s) compared to 20% araticum pulp + 80% water (171,87 mPa.s), the combination of araticum pulp with chickpea extract in the same proportion, 40% araticum pulp to 60% chickpea extract, showed a 3-fold increase (3,959.77 mPa.s) compared to the control, 40% araticum pulp + 60% water (1,307.43 mPa.s). This result suggests an interaction between the components of the araticum pulp and chickpea extract matrices.

The higher viscosity value obtained for formulation F3 (40% araticum pulp + 60% chickpea extract) may be related to the higher concentration of constituents with hydroxyl groups derived from araticum pulp in this formulation. Hydroxyl groups increase the affinity of the compounds to water, acting as natural hydrocolloids, resulting in increased viscosity [62]. In addition, some polysaccharides and proteins are considered hydrocolloids; however, most hydrocolloids are fibers [62, 63]. The formulation F3 had a higher fiber content among formulations, among the polysaccharides already identified in araticum pulp [36, 64], pectin is a hydrocolloid that provides greater firmness in foods [65]. It has also been reported that chickpea proteins can form gels [66]. The proportion of 40% araticum pulp and 60% chickpea extract in F3 was possibly sufficient for an interaction of the components, providing greater viscosity.

It is important to highlight that the viscosity of the formulation F3 (40% araticum pulp + 60% chickpea extract) was similar to that obtained for a commercial yogurt (3,233.80 mPa. s), used as a standard in the analysis, and no additives were added to increase the viscosity of the formulations. This result suggests that formulation F3 can be considered a plant-based dairy alternative. However, further analysis of other physicalproperties of these formulations is necessary to assess whether additives are needed or not.

## Sensory analysis

The results of the sensory evaluation of the different formulations are described in Table 5. Of the participated in the study, of which 68% were women and 32% were men, 80% were between 18–24 years and 20% were between 25–54 years.

The best hedonic classification for color and taste was obtained for formulation F3 (40% araticum pulp + 60% chickpea extract). The odor score was similar among all formulations, while the texture attribute was similar between formulations F2 (30% araticum pulp + 70% chickpea extract) and F3 and higher for these two formulations than formulation F1 (20% araticum pulp + 80% chickpea extract). Despite the high values obtained for color and taste attributes in formulation F3, the overall sensory score of formulations F2 and F3 did not differ; however, it was significantly higher than formulation F1.

In general, hedonic classification varying between the terms "disliked moderately" to "indifferent" was observed for the different aspects evaluated in the formulations, with an exception for the color aspect of the formulation F3 (40% araticum pulp + 60% chickpea extract), which

**Table 5. Sensory evaluation of plant-based dessert formulations using a 9-point hedonic scale and percentage of acceptability.**

| Formulations | Color | % | Taste | % | Odor | % | Texture | % | Overall acceptance | % |
|---|---|---|---|---|---|---|---|---|---|---|
| F1 | 5.0 ± 1.7 [c] | 34.6 | 4.1 ± 2.1 [b] | 25.0 | 5.1 ± 1.7 [a] | 36.5 | 5.2 ± 1.9 [b] | 46.1 | 4.4 ± 2.1 [b] | 36.5 |
| F2 | 5.6 ± 1.6 [b] | 47.1 | 4.5 ± 2.2 [b] | 42.3 | 5.3 ± 1.7 [a] | 40.4 | 5.7 ± 2.0 [a] | 60.5 | 4.9 ± 2.1 [a] | 39.4 |
| F3 | 6.1 ± 1.8 [a] | 59.6 | 4.9 ± 2.3 [a] | 47.1 | 5.5 ± 1.9 [a] | 49.0 | 5.9 ± 1.9 [a] | 61.5 | 5.2 ± 2.1 [a] | 48.0 |

Means followed by the same letter on the columns do not differ statistically by the Tukey test ($P < 0.05$). % = acceptance percentage when considering scores from 6–9 for sensory parameters. Formulation 1 (F1): 20% araticum pulp + 80% chickpea extract, Formulation 2 (F2): 30% araticum pulp + 70% chickpea extract and Formulation 3 (F3): 40% araticum pulp + 60% chickpea extract.

remained within the acceptance scale "liked moderately". For a product to have good acceptability, it is advised to achieve around 70% acceptance in sensory parameters; values below this percentage were obtained for the formulations. The low scores obtained in sensory evaluation may be associated with the striking flavor of araticum pulp, as it is considered an exotic fruit from the Brazilian Cerrado [14, 35]. Schiassi et al. [13], obtained similar results when conducting sensory analysis of fruit juices from the Brazilian Cerrado, and the authors suggested that the flavors of these unconventional fruits are not familiar to tasters, leading to low test scores.

Araticum pulp has a yellow/pink pigment [67], therefore the higher proportion of araticum pulp in the formulation F3 (40% araticum pulp + 60% chickpea extract) intensified its color, making it more attractive to tasters, which explains the greater acceptance for the color aspect in this formulation.

Despite the low hedonic classification of the formulations in the present study, these results were similar to those obtained for other plant-based desserts made with soy, rice, cocoa butter, cashew, coconut, and blueberry, available on the market [68–70]. Rincon et al. [20], demonstrated that the addition of natural flavorings, such as vanilla extract, can be a strategy to enhance the acceptability of plant-based milk. It is possible that such strategies could be applied to plant-based desserts to improve acceptability.

However, the formulation F3 (40% araticum pulp + 60% chickpea extract) presents the highest score in the sensorial aspects and appears to be the most promising formulation for a new plant-based dessert alternative.

## Conclusion

The present study showed that the plant-based dessert elaborated with araticum pulp and chickpea extract might be considered a potential dairy alternative product, as it has a protein content and viscosity similar to yogurt. Furthermore, araticum pulp and chickpea extract plant-based dessert may be a functional food, as it has a high content of phenolic compounds and, consequently, antioxidant activity.

The developed formulation with 40% araticum pulp and 60% chickpea extract (F3) presented the best potential as a plant-based dairy alternative dessert, as it had a viscosity similar to yogurt, acceptable microbiological quality up to 28 days of storage, and the highest antioxidant potential.

Despite the good nutritional value of the araticum pulp and chickpea extract plant-based dessert, the low sensorial acceptability of this dessert suggested that more studies are needed to refine the sensory aspects of these plant-based desserts.

## Author Contributions

**Conceptualization:** Maria Carolina Mesquita, Sandra Fernandes Arruda, Eliana dos Santos Leandro.

**Data curation:** Maria Carolina Mesquita, Márcio Antônio Mendonça, Raquel Braz Assunção Botelho.

**Investigation:** Maria Carolina Mesquita.

**Methodology:** Maria Carolina Mesquita, Sandra Fernandes Arruda, Eliana dos Santos Leandro.

**Project administration:** Maria Carolina Mesquita.

**Resources:** Márcio Antônio Mendonça, Raquel Braz Assunção Botelho, Sandra Fernandes Arruda, Eliana dos Santos Leandro.

**Supervision:** Eliana dos Santos Leandro.

**Validation:** Sandra Fernandes Arruda.

**Visualization:** Maria Carolina Mesquita, Sandra Fernandes Arruda.

**Writing – original draft:** Maria Carolina Mesquita, Eliana dos Santos Leandro.

**Writing – review & editing:** Raquel Braz Assunção Botelho, Sandra Fernandes Arruda.

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
