## [Decision Letter · Decision Letter 0]

14 Jun 2024

PONE-D-24-08819Development of a plant-based dessert using araticum pulp and chickpea extract: physicochemical, microbiological, antioxidant, and sensory characterizationPLOS ONE

Dear Dr. Mesquita,

Thank you for submitting your manuscript to PLOS ONE. After careful consideration, we feel that it has merit but does not fully meet PLOS ONE’s publication criteria as it currently stands. Therefore, we invite you to submit a revised version of the manuscript that addresses the points raised during the review process.

We look forward to receiving your revised manuscript.

Kind regards,

Ghulam Khaliq, PhD

Academic Editor

PLOS ONE

Journal Requirements:

[We thank Coordenação de Aperfeiçoamento de Pessoal de Nível Superior (CAPES) of Brazil for the support with the scholarship, as well as the financial backing provided by the Fundação de Amparo a Pesquisa do Distrito Federal (FAPDF) – Process: 00193-00000104/2019-86.]

 [The author(s) received no specific funding for this work.]

3. Please include your tables as part of your main manuscript and remove the individual files. Please note that supplementary tables (should remain/ be uploaded) as separate ""supporting information"" files".

Reviewers' comments:

Reviewer's Responses to Questions

**Comments to the Author**

1. Is the manuscript technically sound, and do the data support the conclusions?

Reviewer #1: Yes

Reviewer #2: Yes

2. Has the statistical analysis been performed appropriately and rigorously? 

Reviewer #1: Yes

Reviewer #2: N/A

3. Have the authors made all data underlying the findings in their manuscript fully available?

Reviewer #1: Yes

Reviewer #2: Yes

4. Is the manuscript presented in an intelligible fashion and written in standard English?

Reviewer #1: Yes

Reviewer #2: Yes

5. Review Comments to the Author

Reviewer #1: Thank you for providing the opportunity to review the manuscript: Development of a plant-based dessert using araticum pulp and chickpea extract: physicochemical, microbiological, antioxidant, and sensory characterization. The study is quite novel, and relevant considering the demand for plant based foods due to their health benefits. However, some results are missing such as Tables 1 to 4. These results need to be provided in order to make a decision on the manuscript.

Abstract

The authors should improve the presentation of their results. For instance, rather than just stating the measured attributes values, they should indicate the magnitude of increase or decrease. The authors should include the nutritional analysis results to be able compare their product with dairy products such as yogurt.

Material and methods

How were the fruits sanitized and the pulp extracted from the fruit?

The authors should summarize the method they used to prepare chickpea extract and then refer to Figure 1.

Determination of centesimal composition and energy value. What do authors mean by centestimal? Is there not a better, common and simpler word that they can use?

Why did the authors not include a control sample in their microbiological analyses?

Why did the authors not include the relative humidity at 4 °C of their storage facilities. At such low temperature moisture migration between the environment and the product is likely.

The authors should check if 25,400 xg is really xg not rpm

Why did the authors not measure sensory properties such as consistency or stability? I am quite sure the product did not remain consistent and stable throughout the 28 days of storage. This is important if the experimental idea is to be commercialized and if whether additives are needed or not.

Results and discussion

Some results described in the manuscript are missing for instance, Tables 1, 2 and 3.

The authors should improve the discussion of their results. For instance, they should describe the formulations fully or put the % of the ingredients in brackets for easy tracking. e.g., F1 (80:20). This makes it easier for the reader to follow.

Is it not that the pH of the dessert is related to the organic acids in the fruit used? The authors should be able to link the change in pH to TSS results.

Reviewer #2: 1. THE ABSTRAC SUMMARIZES THE CONTEN: 1 ST PARAGARAPH IS TOO LENGTHY

2. AUTHOURS CONTRADICTED THEMSELVES ABOUT FUNDING SOURCE

3 KEY WORDS MUST BE RE-ARRANGED

4.ITRODUCTION[NTRODUCTION SHOULD NOT CONTAIN SO MANY FIGURES OF OTHER PEOPLES WORKS]

5. MATERIALS AND METHODS[what quantity of fruits were purchased, stored for how long before commencement of research, where is the design of the experiment, was the design CRD, as laboratory condition?,

6. RESULST AND DISCUSSION[rewrite 1ST PARAGRAPH, Centesimal composition was NOT STATED IN METODS WHY PRESENT RESULTS, NO TABLE 1,2,3,4,,& 5 SHOWN IN THIS PAPER, just stick to plant based material used in this research chickpea, LL RESULTS WERE COMPARING FORMLATIONS WHY ANTIOXIDANTS ASESSED ONLY ARATICUM PULP

AD ONLY CHICKPEA EXTRACTS?]

6. PLOS authors have the option to publish the peer review history of their article (what does this mean?). If published, this will include your full peer review and any attached files.

Reviewer #1: **Yes: **Dr. Tafadzwa Kaseke

Reviewer #2: No

---

## [Author Response · Author response to Decision Letter 0]

5 Jul 2024

To: Ghulam Khaliq

Academic Editor PLOS ONE

Subject: Revised version of the manuscript PONE-D-24-08819

Dear Ghulam Khaliq,

Thank you by the opportunity to submit a revised version of our manuscript. Below are the responses to each point brought up by the Editor's and reviewers #1 and #2. Some changes made were marked in the “Revised Manuscript with Track Changes” and the answers to reviewer’s questions are presented below. 

Yours sincerely,

*Corresponding author: 

Maria Carolina Mesquita 

Email: maria.carolinams@hotmail.com

Post-Graduate Program in Human Nutrition, Faculty of Health Sciences, Campus Universitário Darcy Ribeiro, University of Brasília, Brasília 70910-900, Brazil Phone: +55 (61) 3107 1634

Journal Requirements:

The manuscript was revised and re-formatted in accordance with Plos One style requirements, including the figures converted to 300 dpi.

[We thank Coordenação de Aperfeiçoamento de Pessoal de Nível Superior (CAPES) of Brazil for the support with the scholarship, as well as the financial backing provided by the Fundação de Amparo a Pesquisa do Distrito Federal (FAPDF) – Process: 00193-00000104/2019-86.]

 [The author(s) received no specific funding for this work.]

We apologize for this mistake. The text related to financing was removed from the manuscript, and a statement was added to the cover letter for later modification: “Inclusion of the Financial Disclosure Statement: This study received financial support from the Fundação de Amparo à Pesquisa do Distrito Federal (FAPDF) - Process: 00193-00000104/2019-86. The funders had no role in study design, data collection and analysis, decision to publish, or preparation of the manuscript.”

3. Please include your tables as part of your main manuscript and remove the individual files. Please note that supplementary tables (should remain/ be uploaded) as separate ""supporting information"" files".

Tables have been included as part of the manuscript and individual files have been removed.

We apologize for the mistake. The files we uploaded as supporting information are documents related to studies involving human (sensory analysis), which were requested at the time of manuscript submission. We have changed the category of these files the "other files" category.

Reviewers' comments:

Reviewer #1: 

Thank you for providing the opportunity to review the manuscript: Development of a plant-based dessert using araticum pulp and chickpea extract: physicochemical, microbiological, antioxidant, and sensory characterization. The study is quite novel, and relevant considering the demand for plant based foods due to their health benefits. However, some results are missing such as Tables 1 to 4. These results need to be provided in order to make a decision on the manuscript.

In the first version of the submitted manuscript, tables were uploaded separately in the submission system, however when the PDF file of the manuscript was built the tables did not appear. We apologize for the error. In this new submission, the tables were included of the manuscript. 

Abstract

The authors should improve the presentation of their results. For instance, rather than just stating the measured attributes values, they should indicate the magnitude of increase or decrease. The authors should include the nutritional analysis results to be able compare their product with dairy products such as yogurt.

The results presentation in the abstract was improved.

Material and methods

How were the fruits sanitized and the pulp extracted from the fruit?

The requested information was included in the text (page 5, line 99 – 103).

The authors should summarize the method they used to prepare chickpea extract and then refer to Figure 1.

The information was summarized in the method section (page 5, line 105 – 111).

Determination of centesimal composition and energy value. What do authors mean by centestimal? Is there not a better, common and simpler word that they can use?

The word 'centesimal' was replaced by 'chemical' throughout the manuscript.

Why did the authors not include a control sample in their microbiological analyses?

We understood that you refer to the control sample as formulations elaborated with different proportions of araticum pulp and water or chickpea extract and water. However, in the microbiological analysis our main objective was to study whether the interaction between these two-plant matrices (araticum and chickpea) would be efficient in maintaining the microbiological stability of plant-based dairy alternative dessert formulations during storage. Considering that both matrices have a high content of polyphenols and that these compounds may present distinct antimicrobial activity in separate matrices compared to the mixture of both matrices, we think that this kind of control would not provide relevant information, since our main purpose was to produce a plant-based dairy alternative dessert based on araticum pulp and chickpea extract.

Why did the authors not include the relative humidity at 4 °C of their storage facilities. At such low temperature moisture migration between the environment and the product is likely.

The relative humidity was included in the material and methods section as suggested. 

The authors should check if 25,400 xg is really xg not rpm.

We are sorry by the mistake. The value was corrected in the text. 

Why did the authors not measure sensory properties such as consistency or stability? I am quite sure the product did not remain consistent and stable throughout the 28 days of storage. This is important if the experimental idea is to be commercialized and if whether additives are needed or not.

We recognize that these parameters are important to ensure that additives are needed or not and for commercialization of the product, however, at this moment only a viscosimeter was available in our faculty. It was included in the discussion section, page 27/ line 554 - 555, the following statement: However, further analysis of other physical properties of these formulations is necessary to assess whether additives are needed or not. 

Results and discussion.

Some results described in the manuscript are missing for instance, Tables 1, 2 and 3.

In the first version of the submitted manuscript, tables were uploaded separately in the submission system, however when the PDF file of the manuscript was built the tables did not appear. We apologize for the error. In this new submission, the tables were included of the manuscript. 

The authors should improve the discussion of their results. For instance, they should describe the formulations fully or put the % of the ingredients in brackets for easy tracking. e.g., F1 (80:20). This makes it easier for the reader to follow.

The percentage of ingredients was included throughout the discussion, as suggested.

Is it not that the pH of the dessert is related to the organic acids in the fruit used? The authors should be able to link the change in pH to TSS results.

It was included in a paragraph in page 16 line 344 - 351.

Reviewer #2: 

1. THE ABSTRAC SUMMARIZES THE CONTEN: 1 ST PARAGARAPH IS TOO LENGTHY

The paragraph has been summarized.

2. AUTHOURS CONTRADICTED THEMSELVES ABOUT FUNDING SOURCE

Sorry for the mistake, the funding source was included in the submission system. 

3 KEY WORDS MUST BE RE-ARRANGED

Keywords have been re-arranged following alphabetic order.

4.ITRODUCTION [INTRODUCTION SHOULD NOT CONTAIN SO MANY FIGURES OF OTHER PEOPLES WORKS]

The sentences were summarized, and some data was deleted to improve this aspect.

5. MATERIALS AND METHODS [what quantity of fruits were purchased, stored for how long before commencement of research, where is the design of the experiment, was the design CRD, as laboratory condition?

The amount of araticum and chickpeas was included in the material and methods section, as well as the storage time of the pulp. Yes, we used the design CRD, and included the experimental design in material and methods section (page 11, lines 245 - 249).

6. RESULST AND DISCUSSION [rewrite 1ST PARAGRAPH, Centesimal composition was NOT STATED IN METODS WHY PRESENT RESULTS, NO TABLE 1,2,3,4,,& 5 SHOWN IN THIS PAPER, just stick to plant based material used in this research chickpea, LL RESULTS WERE COMPARING FORMLATIONS WHY ANTIOXIDANTS ASESSED ONLY ARATICUM PULP AD ONLY CHICKPEA EXTRACTS?]

The 1st paragraph was rewritten according to suggestions.

The description of the chemical composition was provided in the materials and methods (item 2.4, page x, lines x-y) section of the original manuscript. Let me know if we do not understand your comment. 

In the first version of the submitted manuscript, tables were uploaded separately in the submission system, however when the PDF file of the manuscript was built the tables did not appear. We apologize for the error. In this new submission, the tables were included of the manuscript. 

According to the suggestion, it was excluded from the discussion the literature that is related to other plant-based material that is not chickpea. 

The antioxidant potential and phenolic concentration were assessed in the araticum pulp and chickpea extract in an attempt to evaluate if one of these matrices could present different responses in these variables over storage period, since their different chemical composition can influence these responses. As can be seen in table 4 chickpea extract showed better stability during storage compared to araticum pulp which explain the different responses of these variables in the formulations. 

- We leave the term 'untrained participants' as there was no a specific sensory analysis training for participants to evaluate plant-based desserts. Since we conducted an acceptability test, participants were invited randomly and received only the necessary instructions to participate in the research.

- In Figure 2, the asterisk symbol (*) that demonstrated a statistical difference between the formulations were removed, and capital letters were inserted in their place for greater clarity of the results. 

- The declaration of conflict of interest was removed from the manuscript, following Plos One guideline.

---

## [Editor Report · Decision Letter 1]

9 Jul 2024

Development of a plant-based dessert using araticum pulp and chickpea extract: Physicochemical, Microbiological, Antioxidant, and Sensory Characterization

PONE-D-24-08819R1

Dear Dr. Maria Carolina Mesquita,

We’re pleased to inform you that your manuscript has been judged scientifically suitable for publication and will be formally accepted for publication once it meets all outstanding technical requirements.

Kind regards,

Ghulam Khaliq, PhD

Academic Editor

PLOS ONE

Additional Editor Comments (optional):

The manuscript could be accepted after revision.
---

## [Editor Report · Acceptance letter]

15 Jul 2024

PONE-D-24-08819R1 

PLOS ONE

Dear Dr. Mesquita, 

I'm pleased to inform you that your manuscript has been deemed suitable for publication in PLOS ONE. Congratulations! Your manuscript is now being handed over to our production team.

Kind regards, 

on behalf of

Dr. Ghulam Khaliq 

Academic Editor

PLOS ONE